# A Nano-Emulsion Platform Functionalized with a Fully Human scFv-Fc Antibody for Atheroma Targeting: Towards a Theranostic Approach to Atherosclerosis

**DOI:** 10.3390/ijms22105188

**Published:** 2021-05-14

**Authors:** Samuel Bonnet, Geoffrey Prévot, Stéphane Mornet, Marie-Josée Jacobin-Valat, Yannick Mousli, Audrey Hemadou, Mathieu Duttine, Aurélien Trotier, Stéphane Sanchez, Martine Duonor-Cérutti, Sylvie Crauste-Manciet, Gisèle Clofent-Sanchez

**Affiliations:** 1Centre de Résonance Magnétique des Systèmes Biologiques, CNRS UMR 5536, Université de Bordeaux, CRMSB, 33076 Bordeaux, France; marie-josee.jacobin-valat@rmsb.u-bordeaux.fr (M.-J.J.-V.); audrey.hemadou@orange.fr (A.H.); aurelien.trotier@rmsb.u-bordeaux.fr (A.T.); stephane.sanchez@rmsb.u-bordeaux.fr (S.S.); gisele.clofent-sanchez@rmsb.u-bordeaux.fr (G.C.-S.); 2Institut de Chimie de la Matière Condensée, CNRS UMR 5026, Université de Bordeaux, Bordeaux INP, ICMCB, 33600 Pessac, France; Stephane.Mornet@icmcb.cnrs.fr (S.M.); Mathieu.Duttine@icmcb.cnrs.fr (M.D.); 3ARNA, ARN, Régulations Naturelle et Artificielle, ChemBioPharm, INSERM U1212, CNRS UMR 5320, Université de Bordeaux, 33076 Bordeaux, France; geoffrey.prevot@gmail.com (G.P.); ymousli33@gmail.com (Y.M.); sylvie.crauste-manciet@u-bordeaux.fr (S.C.-M.); 4CNRS UPS 3044, Baculovirus et Thérapie, 30380 Saint-Christol-lès-Alès, France; Martine.CERUTTI@cnrs.fr

**Keywords:** atherosclerosis, nano-emulsion, magnetic resonance imaging, stealth, human antibody

## Abstract

Atherosclerosis is at the onset of the cardiovascular diseases that are among the leading causes of death worldwide. Currently, high-risk plaques, also called vulnerable atheromatous plaques, remain often undiagnosed until the occurrence of severe complications, such as stroke or myocardial infarction. Molecular imaging agents that target high-risk atheromatous lesions could greatly improve the diagnosis of atherosclerosis by identifying sites of high disease activity. Moreover, a “theranostic approach” that combines molecular imaging agents (for diagnosis) and therapeutic molecules would be of great value for the local management of atheromatous plaques. The aim of this study was to develop and characterize an innovative theranostic tool for atherosclerosis. We engineered oil-in-water nano-emulsions (NEs) loaded with superparamagnetic iron oxide (SPIO) nanoparticles for magnetic resonance imaging (MRI) purposes. Dynamic MRI showed that NE-SPIO nanoparticles decorated with a polyethylene glycol (PEG) layer reduced their liver uptake and extended their half-life. Next, the NE-SPIO-PEG formulation was functionalized with a fully human scFv-Fc antibody (P3) recognizing galectin 3, an atherosclerosis biomarker. The P3-functionalized formulation targeted atheromatous plaques, as demonstrated in an immunohistochemistry analyses of mouse aorta and human artery sections and in an *Apoe*^−/−^ mouse model of atherosclerosis. Moreover, the formulation was loaded with SPIO nanoparticles and/or alpha-tocopherol to be used as a theranostic tool for atherosclerosis imaging (SPIO) and for delivery of drugs that reduce oxidation (here, alpha-tocopherol) in atheromatous plaques. This study paves the way to non-invasive targeted imaging of atherosclerosis and synergistic therapeutic applications.

## 1. Introduction

Atherosclerosis is characterized by the development of lipid-rich plaques, called atheromatous plaques, in the artery wall [1]. Atheromatous plaques can be classified into two types: stable plaques and vulnerable plaques [2]. Stable plaques are usually rich in extracellular matrix and smooth muscle cells that maintain the integrity of these fibrous plaques for years. Conversely, vulnerable plaques are rich in macrophages and inflammatory cells that make them prone to rupture, leading to cardiovascular complications [3,4,5].

Currently, angiography is normally used for imaging peripheral arterial disease [6]. However, it gives information only on vessel lumen reduction (stenosis) but not on the plaque morphology and risk of rupture [7]. Moreover, in two-thirds of ruptured plaques, stenosis is insignificant on angiograms [8,9]. Recently, other imaging modalities have emerged. For instance, intravascular ultrasound and optical coherence tomography [10,11,12,13] provide information on plaque morphology but are invasive procedures. Other non-invasive imaging strategies might offer new opportunities for atheroma diagnosis [14,15], particularly magnetic resonance imaging (MRI), which combines excellent soft-tissue contrast, good resolution, and absence of exposure to ionizing radiation [16,17,18,19].

MRI with highly specific targeting probes, in which contrast agents are conjugated to antibodies against molecular components of the atheromatous plaque, might allow the contrast agents to be directed specifically to the lesions. Molecular imaging of vulnerable plaques is of utmost interest because plaque composition might contribute to plaque rupture more than artery narrowing. The aim of this study was to develop an improved targeted contrast agent using nano-emulsions (NE) and the first human antibody (HuAb) against galectin-3 (HuAb P3, WO2019068863A1). Indeed, previous work demonstrated that oil-in-water NEs loaded with superparamagnetic iron oxide (SPIO) nanoparticles generated an accurate MRI signal, making them highly suitable as imaging agents [20,21]. The P3 antibody was chosen because galectin-3 is strongly expressed by the TREM2-positive foamy macrophage subset [22], which has been recently identified by single-cell RNA sequencing as the main immune cell subset in atherosclerosis. Importantly, the TREM2-positive subset endowed with specialized functions in lipid metabolism and catabolism was almost exclusively detectable in atherosclerotic aortas and present at different time points of lesion formation [23]. Moreover, using a HuAb decreases the potential immunogenicity in clinical settings. Before conjugation to the P3 HuAb, the NE surface was decorated with polyethylene glycol (PEG), the most common method to reduce clearance from the blood circulation [24]. PEG macromolecules create a protective hydrophilic layer around nanoparticles that can repel binding by opsonin proteins (i.e., opsonization) [25] and increase their half-life in the blood [26]. PEGylation of liposomes [27], micelles [28], and nanoparticles [29] has been widely investigated, but only a few works have focused on NE surface modification with PEG. Hak et al. [30] studied the impact of PEG surface density on NE half-life by blood sampling at different time points after administration in mice. Here, the effect of NE surface modification by PEG layers of different molecular weights (PEG_2000_ and/or PEG_3400_) on liver uptake was studied by dynamic MRI. The HuAb P3 engineered via the single chain fragment variable (scFv)-Fc format was conjugated to the formulation with the lowest liver uptake and a longer half-life according to previous procedures [31].

Finally, theranostic PEGylated NEs loaded with SPIO nanoparticles and alpha-tocopherol were developed. Oral supplementation with alpha-tocopherol failed to show a clear benefit on the reduction of cardiovascular events in clinical trials [32]. Indeed, systemic administration does not allow one to reach the critical active concentration of the specific antioxidant at key sites [33]. In this study, we aimed to tackle this issue by proposing a targeted drug delivery strategy that could overcome the failure of specific tissue concentration of orally administered antioxidants. The antioxidant properties of PEGylated NEs loaded with SPIO nanoparticles and alpha-tocopherol were assessed as a multi-modal tool for atherosclerosis imaging and treatment.

## 2. Results

### 2.1. PEGylated NE Formulations and Characterization

Formulations were prepared using Miglyol 840, a neutral pharmaceutical oil based on propylene glycol diether of C8 and C10 saturated plant fatty acids. Two surfactants were used to stabilize the oily droplets, Tween 80 and Lipoid E80. The NE formulations were loaded with maghemite-based SPIO nanoparticles to develop T_2_*-shortening MR contrast agents with relaxivity values close to the gold standard, as previously described [20] ($r_{2}^{*}$ = 42 to 45 mM^−1^∙s^−1^ and r_1_ < 0.1 mM^−1^∙s^−1^). To produce stealth NEs with limited liver uptake and, consequently, a longer half-life in the blood circulation, the droplet surface was PEGylated. PEG creates a hydrophilic and biocompatible layer that limits opsonin adsorption [34,35] and non-specific cellular uptake compared with unmodified carriers. Several therapeutic strategies using PEG have been approved by the Food and Drug Administration (FDA) [25]. In this study, two lipid–PEG combinations with different molecular weight and functionalization were used: DSPE–PEG_2000_ and DSPE–PEG_3400_–maleimide. The maleimide linker allows the conjugation of the HuAb P3. The two lipid–PEG combinations were added at the same molar concentration (5 µmol/mL) to the oily phase before the phase inversion step. Some formulations included only lipid–PEG_2000_ (#2), only lipid–PEG_3400_–maleimide (#3), or a mixture of both (#4). Non-PEGylated NEs (#1) were used as controls.

Decoration of the droplet surface with PEG increased the hydrodynamic diameter from 175.8 nm for NE (#1) to 190.9 nm, 197.2 nm, and 191.0 nm, for NE–PEG_2000_ (#2), NE–PEG_3400_–maleimide (#3), and NE–PEG_2000/3400_–maleimide (#4), respectively. The size of all NE formulations remained in the submicronic range and their polydispersity index (PdI) was <0.2 (monodisperse samples) (Table 1). Thus, the formulation diameter remained smaller than the tiniest blood vessel, preventing their occlusion [36]. Moreover, the nature of the lipids associated with NE plasticity might favor deeper tissue penetration and biological barrier crossing [37].

SPIO nanoparticle inclusion in the oily droplets was confirmed by transmission electron microscopy (TEM) analysis (Appendix A). Nanoparticle tracking analysis (NTA) was used to determine the size and number of submicron particles. The size distribution was consistent with the dynamic light scattering (DLS) analysis results and the droplet number was 5.75 × 10^13^ ± 3.66 × 10^12^ droplets per mL. To our knowledge, this is the first time oily droplet number has been determined to control the theoretical antibody:PEGylated NE ratio.

### 2.2. Stealthy Features of PEGylated NEs

Among the several promising new drug delivery systems, NEs are an advanced technology used to carry molecules to a specific site, and several NE formulations are already used in clinics [38,39,40]. Before improving NE targeting thanks to an antibody conjugated to its surface, it was important to characterize the in vivo stealthy feature of each NE formulation. Stealth is a parameter directly related to the half-life in the bloodstream. Here, the stealthy behavior of three PEGylated NE formulations ((#2), (#3), and (#4)) and of non-PEGylated NE ((#1); control) was studied by dynamic MRI after NE injection at the same iron concentration (3 mg/kg bodyweight) in the tail vein of C57BL/6 mice. The iron concentration of 3 mg/kg was chosen based on guidelines and doses used in the literature for ferumoxytol, approved by the FDA for anemia treatment. The dose of 3 mg/kg of ferumoxytol seems to be well tolerated for MRI-based diagnostic imaging without serious adverse events, according to a recent multi-centric study that underlined the positive safety profile [41]. Liver uptake was monitored continuously by MRI for about 10 min before and up to 50 min after injection (Figure 1).

Depending on the lipid–PEG molecular weight, the formulations displayed different liver uptake patterns. After injection, the non-PEGylated formulation (#1) rapidly accumulated in the liver and the signal was saturated at 2 min after injection, confirming its less stealthy properties. Such rapid NE clearance is consistent with literature data showing that typically up to 90% of the formulation is taken up by the liver within 5 min [36]. NEs decorated with lipid–PEG_2000_ (#2) or the mixture of two PEGs (#4) showed similar clearance profiles, with a rapid liver uptake after injection, although it was slower compared with (#1). NEs functionalized with lipid–PEG_3400_–maleimide (#3) showed the lowest liver uptake at 50 min post-injection, and therefore the best stealth profile. The MR signal in the kidneys remained stable throughout the in vivo experiments, with all tested formulations.

On the basis of these in vivo results, the NE–PEG_3400_–maleimide (#3) formulation was chosen, due to its having the lowest liver uptake, for conjugation with scFv-Fc-2Cys P3 HuAb to obtain NE-P3 (#5). The next experiments focused on the characterization of NE-P3 (#5).

### 2.3. Bio-Conjugation with P3 HuAb and In Vivo Clearance of NE-P3

Conjugation of HuAb P3 to the NE–PEG_3400_–maleimide surface (NE-P3, (#5)) induced an increase in NE diameter, assessed by DLS (199.5 ± 2.3), as previously reported [20]. The formulation of NE-P3 (#5) was monodisperse (PdI = 0.167) and its size was still in the submicronic range, with a diameter similar to that of commercial NEs used for human parenteral nutrition [36]. To determine whether the addition of antibody affected the formulation’s stealthy properties, the in vivo clearance of the NE-P3 (#5) formulation was evaluated by dynamic MRI monitoring of liver uptake for 24 h after injection in the tail vein of one *Apoe*^−/−^ mouse (Figure 2). This experiment allowed calculation of the NE-P3 (#5) half-life (103 min) in blood (Table 2).

After calculation of NE-P3’s (#5) half-life in the bloodstream, its targeting efficiency was assessed in vitro, in vivo, and ex vivo (see below).

### 2.4. In Vitro Atheroma Targeting

First, NE-P3’s (#5) ability to specifically target atheromatous plaques was tested by immunohistochemistry (IHC) using aorta tissue sections from hypercholesterolemic *Apoe*^−/−^ mice. A strong positive signal was observed only in aorta sections incubated with NE-P3 (#5) (Figure 3(Ab)), but not with NE–PEG_3400_–maleimide (#3) (without antibody) (Figure 3(Ad)). The uniform intense signal obtained with NE-P3 (#5) compared with the free P3 antibody, used at the same concentration, (compare Figure 3(Aa,Ab)) is certainly due to the ratio of 14 antibody molecules per PEGylated NE droplet used for the bio-conjugation that contributed to increase P3’s binding avidity.

Next, NE-P3 (#5) and PEG_3400_-maleimide (#3) were tested on human endarterectomy samples (Figure 3B). As before, a strong signal was observed only in atheromatous plaques incubated with NE-P3 (#5) (Figure 3(Bb)) but not with NE-PEG_3400_-maleimide (#3) (Figure 3(Bd)). The faint signal observed with NE–PEG_3400_–maleimide (#3) was mostly due to human immunoglobulins in atheromatous plaques. Moreover, the NE-P3 (#5) signal was stronger than that of the unconjugated P3 antibody (positive control) (Figure 3(Ba)), as observed in the mouse sections. These findings indicate that NE-P3 (#5) efficiently targets atheromatous lesions in mouse aortas (pre-clinical model) and also in human endarterectomy samples (for future translational assays). These results underline NE-P3’s (#5) potential as a new molecular contrast agent to target atherosclerosis by binding to galectin-3.

### 2.5. Preliminary Data on In Vivo Atheroma Targeting by NE-P3

To determine whether NE-P3 (#5) can also target atheromatous plaques in vivo, this formulation was injected in the tail vein of one *Apoe*^−/−^ mouse and atheroma targeting was monitored by MRI. The T_2_* relaxation maps of atheromatous plaques were then computed from a multi-slice (RF)-spoiled gradient echo sequence (multi-echo GRE) obtained at 4.7 T at different time points: before (baseline), 7 h, and 24 h after NE-P3 (#5) injection. Aorta segmentation and T_2_* mapping were performed concomitantly by two skilled experimenters. The slice-by-slice diagrams representing the T_2_* mean values (in ms) at baseline, 7 h and 24 h post-injection calculated by the two experimenters (Figure 4A) showed a slice-by-slice decrease in the T_2_* mean values, with differences between slices particularly at 7 h after injection. As the plaque characteristics (e.g., size, permeability, components) can change along the aorta, it is not surprising to observe differences in T_2_* decrease from one slice to another. Figure 4B–D shows the segmented T_2_* maps of the abdominal atheromatous plaque for the same five adjacent slices before, at 7 h, and at 24 h after injection, respectively. For each slice, the T_2_* map was overlaid on its corresponding magnitude image (i.e., the last panel, in the lower right corner, for each time point in Figure 4B–D). Comparisons of the data at the three time points (Figure 4B–D) highlighted the decrease in T_2_* values at 7 h post-injection and their increase again (but still lower than at baseline) at 24 h post-injection. These first results showed that NE-P3 (#5) could be used for in vivo targeting of atheromatous plaques and indicated that the in vivo behavior of NE-P3 (#5) must be accurately monitored over time to determine the optimal imaging window.

### 2.6. Ex Vivo Analyses of Isolated Aorta Samples

Ex vivo studies were then carried out after in vivo MRI imaging to confirm the presence of the targeted contrast agent in the aorta using two methods: MRI (to visualize the areas with a decreased T_2_* signal) and Electron Spin Resonance (ESR, to quantify the iron present in the aorta).

#### 2.6.1. Ex Vivo MRI

Ex vivo MRI scans of the aorta embedded in agarose gel after isolation from one *Apoe*^−/−^ mouse following in vivo injection of NE-P3 (#5) in the tail vein highlighted the intimal thickening characteristic of an atherosclerotic plaque. A drop in the MR signal due to the T_2_* effect characterized the accumulation of the iron oxide-based targeted contrast agent in the atherosclerotic lesions (black arrows in Figure 5).

#### 2.6.2. Iron Oxide Accumulation Assessment by ESR

Iron quantification by ESR in the same isolated aorta corroborated the detection of iron accumulation in the atheromatous plaque due to NE-P3 (#5) targeting. After injection, the broad and intense ESR signal detected at about 320 mT (g = 2.12) with a peak-to-peak line width of about 80 mT (Figure 6) was due to the presence of Fe_2_O_3_ nanoparticles at an estimated concentration of 1.7 ± 0.4 × 10^−8^ mol/g in the analyzed tissue.

### 2.7. Antioxidant Alpha-Tocopherol-Containing NEs for a Theranostic Approach: Formulation, Characterization, and Activity

For this theranostic approach, NE samples decorated with PEG_3400_–maleimide (4.4 µmol∙mL^−1^) were combined with alpha-tocopherol and SPIO (12.5 µmol∙mL^−1^) in the oily phase (NE–PEG_3400_–maleimide–SPIO–tocopherol, (#10)) (Table 3). To determine the antioxidant or prooxidant contribution of each component of this formulation, four other formulations were assessed as controls (Table 3). All formulations were in the submicronic size range and monodisperse (PdI < 0.3) (Table 4).

First, the antioxidant properties of these NE formulations were assessed by comparing their ability to increase the time to produce the hemolysis of 50% of red blood cells (T50% hemolysis, in minutes) after the initiation of free radical attack (Figure 7).

The results indicated that NEs without alpha-tocopherol (#6) and NE–PEG_3400_–maleimide–SPIO (#8)) had a slight prooxidant effect. When used at the concentration of 1000 mg/L, they reduced the T50% hemolysis by 10.95% and 10.66%, respectively, compared with the reference sample. Conversely, alpha-tocopherol-containing formulations ((#7), (#9), and (#10)) at the concentration of 1000 mg/L) significantly and similarly increased the T50% hemolysis by up to 104.63%. However, the presence of iron oxide nanoparticles (#10) slightly hindered alpha-tocopherol’s antioxidant activity. Furthermore, when the results were expressed as Trolox equivalents (Figure 8), which is a water-soluble analog of Vitamin E commonly used in biological and biochemical applications as an antioxidant reference, the Trolox equivalent of 1 g of alpha-tocopherol-containing NEs ranged from 39.88 mg to 44.62 mg. These values were close to the amount of alpha-tocopherol loaded in the NEs (50 mg), thus proving that the formulation does not impair or hinder alpha-tocopherol’s antioxidant activity.

## 3. Discussion

As cardiovascular complications caused by atherosclerosis are the leading cause of death in Western countries, it is crucial to develop tools for the early detection of vulnerable atheromatous plaques. In this context, the combination of nanotechnology and molecular imaging is a promising non-invasive strategy for detecting unstable plaques. Providing multi-territorial imaging of the atherosclerotic disease burden is now seen as necessary for a comprehensive patient assessment. Indeed, the mechanism of “vulnerable” plaque rupture is clearly more complex than initially assumed, and focusing only on the treatment of a single atherosclerotic plaque may not necessarily lead to a survival advantage. Proof of this is a recent systematic review and meta-analysis showing that there is no survival benefit of revascularization among patients with stable ischemic heart disease [42]. Numerous clinical investigations have demonstrated that many plaques rupture without clinical symptoms [43]. Moreover, plaque morphology changes over a few months, highlighting the necessity of longitudinal imaging studies. Our strategy is in keeping with new concepts indicating that risk would be more strongly predicted by detecting the total atheromatous burden of the arterial tree in the whole body [44]. The development of non-invasive molecular MRI imaging modality using contrast agents functionalized with antibodies capable of detecting high-risk plaques would allow longitudinal studies and assess the dynamic nature of atherosclerotic disease for a comprehensive approach to the atheroma burden in the “vulnerable” patient.

Due to their high magnetic susceptibility and nanometric size, SPIO nanoparticles have been extensively investigated as MRI contrast agents [5]. However, they must be loaded into a nanocarrier to improve their biocompatibility and half-life. In this work, SPIO nanoparticles made hydrophobic by an oleic acid coating were loaded inside the oily core of PEGylated NEs functionalized with the first anti-galectin-3 human antibody for specific atheroma targeting. When classical iron oxide-based contrast agents are used, the MR signal changes observed during their accumulation in tissues are weighted both by the $R_{2}^{*}$ and R_1_ effects. In the environment of oily droplets, SPIO nanoparticles do not have any access to water and, therefore, their tissue accumulation mainly affects the local $R_{2}^{*}$ relaxation rate (and not the R_1_). We took advantage of the “pure” $R_{2}^{*}$-increasing effect obtained in oily droplets to estimate the blood half-life of each tested NE formulation.

NEs have been in medical use for more than five decades as a parenteral nutrition system for patients who cannot be fed orally (e.g., Intralipid, approved in Europe in 1962) [36]. The huge potential of NEs as drug delivery systems is currently unexploited despite the advantages compared with other nanocarriers [45,46]. To target atheroma with NEs, their half-life must be increased. This is commonly achieved by decorating the droplet surface with PEG chains (i.e., PEGylation) to delay opsonization. PEG has the advantages of being FDA-approved, soluble in hydrophilic and hydrophobic phases, non-toxic, and non-immunogenic [47]. PEG is also available in different molecular weights. For long-term circulation, PEG chains with a molecular weight of at least 2000 must be used [45,48,49,50].

To our knowledge, only a few studies have investigated how to prolong the half-life of PEGylated NEs [30,51,52]. Cheng et al. studied how different molecular weights and different concentrations of lipid–PEG affected NE size but did not analyze their pharmacokinetic profiles [52]. Hak et al. studied the influence of lipid–PEG_2000_ density on NE pharmacokinetics by fluorescence analysis of animal serum samples [53]. In all these studies, NE pharmacokinetics were assessed by blood sampling, while our approach relies on the in vivo determination of NE uptake by the liver using MRI. In our study, the kinetics of four NE formulations were compared. The mean NE diameters tended to increase with the increase in PEG molecular weight, confirming PEG’s brush-like conformation, whereas a mushroom-like conformation would result in a decrease in NE diameter [51,52,53,54]. Dynamic MRI is a powerful tool to rapidly follow, in a longitudinal manner, the stealth properties of nanocarriers over time, in contrast to classical methodologies that rely on blood sampling at defined time points.

The dynamic MRI approach showed that the half-life of NE–PEG_3400_–maleimide (#3) was significantly increased compared with the other formulations. Stealth properties are important for extending the half-life and consequently improving the targeting efficacy. To develop an optimized tool for molecular imaging of atherosclerosis, an atheroma-specific HuAb, P3, was chosen for conjugation to the NE–PEG formulation with the longest half-life (NE–PEG_3400_–maleimide (#3)). P3 specifically targets galectin-3, a protein that has been highlighted in recent studies as a new atherosclerosis biomarker [55]. P3 was discovered by in vivo phage-display selection in an animal model of atherosclerosis using a human scFv library [56,57]. Its ability to target galectin-3 within atherosclerotic lesions has been demonstrated in vitro and ex vivo (patent WO2019068863A1). P3 variable domains were engineered in the scFv-Fc format with a 2Cys tag for site-specific conjugation to NE–PEG_3400_–maleimide (#3) using thiol–maleimide “click” chemistry. Site-specific conjugation has many advantages; particularly, it maintains bioreactivity and can be achieved on the Fc fragment. A theoretical ratio of 14 antibodies per droplet was chosen for efficient molecular targeting, without denaturing the system. Indeed, previous work on SPIO nanoparticles showed that too high a ratio causes object flocculation. Compared with other synthetic nanocarriers, such as SPIO and ultrasmall paramagnetic iron oxide (USPIO) nanoparticles, which are rigid objects, oily droplets have a soft consistency and can efficiently cross the vascular fenestrae of the impaired endothelium and penetrate into the targeted lesions, even if their size is >25 nm. Other groups have demonstrated that lipid-based formulations up to 200 nm in diameter can enter a plaque [58,59,60]. The preliminary results of the assay testing whether in vivo NE–PEG_3400_–maleimide-P3 (#5) can target the atheromatous plaques located in the abdominal aorta of one *Apoe*^−/−^ atherosclerotic mouse are sufficiently encouraging to justify additional in vivo MRI studies to determine the best imaging window. To this end, the contrast agent’s accumulation must be accurately monitored at different time points: directly after injection and up to 48 h post-injection.

In this work, IHC experiments showed that the P3 HuAb can recognize galectin-3 in the *Apoe*^−/−^ mouse aorta and also in human atheroma samples. It should be noted that the ligands used in the more recent studies for functionalizing nanoparticles, micelles, or liposomes tend to be non-immunogenic molecules that cross-react with different species, including humans [61,62,63]. This could facilitate the translation from pre-clinical to clinical studies. Additionally, if the developed diagnostic agents reach the clinics, the use of human antibodies will limit the risk of immunogenicity, thus saving the time required for the humanization of murine antibodies.

Finally, the use of such NEs as a potential theranostic tool was demonstrated using alpha-tocopherol as an antioxidant agent that could reduce the proliferation of radical species inside the atheromatous plaque, thus potentially decreasing the risk of rupture. The “Kit Radicaux Libre” test (KRL test) clearly showed that alpha-tocopherol-containing NEs display antioxidant properties (not observed with NEs without alpha-tocopherol), and also that the full potential of the encapsulated antioxidant agent is available to counteract a free radical attack. Indeed, the antioxidant activity expressed in Trolox equivalents was close to the amount of antioxidant agent loaded inside the NEs, demonstrating that all the alpha-tocopherol contained in the formulation was used in the test. However, it is worth noting that adding SPIO nanoparticles may have a slight prooxidant effect. Moreover, a limitation of this study is the lack of a toxicity assessment. If we want to further develop this theranostic approach using NE formulations that include SPIO nanoparticles, this point should be addressed in priority, although, here, NE-P3 was injected at 3 mg/kg, which was lower than the dose of ferumoxytol, an intravenous iron preparation, used in clinics to treat iron deficiencies. Overall, these results encourage us to develop a complex multi-modal theranostic approach for the diagnosis and treatment of atherosclerosis that could be tested longitudinally in small animal models of this disease in pre-clinical studies. The full potential of the theranostic approach would need to be tested in future in vivo studies using the NE–PEG_3400_–maleimide formulation loaded with SPIO and tocopherol, and functionalized with the scFv-Fc P3 human antibody.

More generally, this work shows the development of a nanomedicine platform that could be advantageously used with stronger reductant molecules chemically derived from tocopherol [33] or other payloads such as prostacycline for its anti-aggregant properties, which are of high value in atherothrombosis [64]. It could also be used with alternative HuAbs. An anti-platelet HuAb under the same format including cysteines for site-specific functionalization has been successfully grafted on the same platform [20]. The SPIO-loaded NE–PEG_3400_–maleimide formulation could thus be adapted to other pathologies, just by changing the active principle and the targeting HuAb.

## 4. Materials and Methods

### 4.1. Materials

Purified Miglyol 840 (oil phase) was kindly provided by IOI OLEO GmbH (Hamburg, Germany). Egg lecithin containing 82.3% phosphatidylcholine (Lipoid E80) and N-(carbonyl-methoxypolyethylenglycol-2000)-1,2-distearoyl-sn-glycero-3-phosphoethanolamine (DSPE–PEG_2000_) were provided by Lipoid GmbH (Ludwigshafen, Germany); polysorbate 80 (Tween 80) was purchased from SEPPIC (Paris, France). The hetero-bifunctional linker 1,2-distearoyl-sn-glycero-3-phosphoethanolamine-N-poly(ethylene glycol)-maleimide (DSPE–PEG_3400_–maleimide) was purchased from Laysan Bio, Inc. (Arab, AL 35016, USA). SPIO nanoparticles were synthesized and made hydrophobic by following previously described procedures [65,66,67]. Tris(2-carboxyethyl)phosphine hydrochloride (TCEP; ≥98%), hydrochloric acid, formic acid, the PBS buffer, sodium hydroxide, and IgG were bought from Sigma-Aldrich (St. Louis, MO 63178, USA). The PBS–heparin solution was from Sanofi Aventis (Vitry-sur-Seine, France). Glycerol was purchased from Cooperation Pharmaceutique Française (Melun, France). MACS Cell Separation Columns were from Miltenyi Biotec (Bergisch Gladbach, Germany).

### 4.2. NE Formulations

NEs were formulated as previously described [20]. Briefly, 12 mg of Lipoid E80 was dispersed in 200 mg of Miglyol 840 (IOI Oleo GmbH, Hamburg, Germany) by heating, followed by addition of 12.5 µmol/mL SPIO nanoparticles. The aqueous phase was a dispersion of 25 mg Tween 80 in 800 mg Milli-Q water. For NE PEGylation, lipid–PEG at different molecular weights (DSPE–PEG_2000_ and/or DSPE–PEG_3400_-maleimide) was added at 5 µmol/mL. A molar ratio of 3:1 was used for the PEG_2000_/PEG_3400_–maleimide mixture. The composition of the different NE formulations is described in Table 5. NE-P3 was obtained after bio-conjugation with HuAb P3, as described in Section 4.4.

Alpha-tocopherol-containing NEs were obtained as before, except that the oily phase included 50 mg of alpha-tocopherol and 150 mg of Miglyol 840. The compositions of the different theranostic NE formulations are described in Table 3.

All mixtures were emulsified by phase inversion and homogenized by sonication (Sonic Vibra Cell–VC 250 set at 70% and output 7; Sonics & Materials Inc, Newtown, CT 06470, USA) to obtain oil droplets in the submicron size range. Before the in vitro and in vivo experiments, the pH of the different formulations was adjusted to the physiological value using 0.1 N sodium hydroxide, and 2% glycerol was added to adjust osmolality.

### 4.3. NE Characterization

The physical characteristics of the NE formulations were assessed by DLS, Zeta potential measurement, and TEM. The hydrodynamic size was determined using a DLS device (Zetasizer Nano ZS; Malvern Instruments, Malvern, UK) and with the NEs diluted to 1:1000 (*v*/*v*) (mean of 3 independent measurements performed at 25 °C). The Zeta potential was measured using a Zetasizer Nano ZS device coupled to a Folded Capillary Cell (DTS1060) from Malvern Instruments. Oily droplets were quantified by NTA using a NanoSight NS300 instrument (Malvern Instruments). A Hitachi H7650 transmission electron microscope linked to an ORIUS SC1000 11MPX (Gatan Inc., Pleasanton, CA, USA) camera run by Digital Micrograph (Gatan Inc.) was used to study the NE samples (1:50 dilution, *v*/*v*) transferred to a carbon-coated copper grid. Iron concentration was quantified by UV spectrometry as previously described [66].

### 4.4. P3 Antibody Bio-Conjugation to NEs

To achieve optimal targeting of atheromatous plaques, the HuAb P3 was engineered in the single chain fragment variable (scFv)-Fc format with a 2-cysteine tag (ScFv-Fc-2Cys) for site-specific conjugation to the NE’s surface. The production of antibodies with this cysteine tag has been described previously for the TEG4 antibody [31]. Before bio-conjugation to NE–PEG_3400_–maleimide (#3), the thiol groups on the cysteine tag of ScFv-Fc-2Cys P3 were activated using TCEP (20 mol per mol of P3). Antibody conjugation to the thiol-reactive maleimide of NE–PEG_3400_–maleimide (#3) to obtain NE-P3 (#5) was performed overnight with a theoretical ratio of 14 antibody molecules per NE droplet. Unconjugated antibodies were removed using a magnetic sorting column (MACS Cell Separation Columns), as previously described [31].

### 4.5. In Vitro Immunoreactivity Analysis by IHC

NE-P3 immunoreactivity was assessed by IHC using paraffin-embedded tissue sections of mouse and human atheromatous plaques. Human biopsies were provided by the vascular and general surgery service of the Pellegrin academic hospital, Bordeaux, France. Samples were from patients who underwent endarterectomy after an acute vascular event. All clinical interventions were carried out at Pellegrin hospital and the use of human samples for research was approved by the Bordeaux CPP ethics committee (Comité de Protection des Personnes Sud-Ouest et Outre Mer) and by the French Research Ministry (Authorization number DC-2016-2724). The CPP committee waived the need for the patients’ written consent because surgical waste no longer attached to the person is considered res nullius. Nevertheless, the patients were informed by the clinicians; if they did not express their opposition to research, de-identified samples were immediately processed and embedded in paraffin. IHC experiments were performed as previously described [31]. Briefly, after deparaffinization, heat-induced epitope retrieval and non-specific interaction blocking, as described in [56], aorta sections were incubated with NE-P3 ((#5), 53 µg/mL of P3 antibody, 158 mg/L of Fe), unconjugated NE–PEG_3400_–maleimide ((#3), 158 mg/L of Fe), ScFv-Fc-2Cys P3 HuAb (53 µg/L) (positive control), or the diluent alone (negative control) overnight. This was followed by incubation with the secondary HRP-conjugated goat anti-human antibody (Fcγ-specific; 1:1000 (*v*/*v*) (Jackson ImmunoResearch; West Grove, PA, USA), and antibody binding was revealed using the Dako Liquid DAB+ Substrate Chromogen System (Agilent Technologies Inc, Santa Clara, CA, USA).

### 4.6. In Vivo Experimental Animal Model

Six-week-old female C57BL/6 mice (weighing 17 to 18 g) and 8- to 12-month-old JAX *Apoe*^−/−^ mice (28 to 32 g) were purchased from Charles River Laboratories (Saint Germain Nuelles, France) and housed under a 12 h light/dark cycle with food and water provided ad libitum. *Apoe*^−/−^ mice were fed a high-cholesterol diet (0.15% cholesterol) for 21 weeks to allow the development of atherosclerotic lesions. Experimental animals were cared for in accordance with institutional guidelines, and they were acclimatized for at least 7 days before the initiation of any experiment. All preclinical experiments described in this publication were approved by the Animal Care and Use Committee of Bordeaux, France (N°50120192-A).

### 4.7. Magnetic Resonance Imaging

All MR experiments and relaxometry measurements were performed at 37 °C using a 4.7-Tesla Bruker Biospec System (Ettlingen, Germany) equipped with a gradient system with a maximum strength of 660 mT/m and a 110 μs rise time.

#### 4.7.1. In Vivo Assessment of Stealthy Features by Dynamic MRI

Dynamic MRI was performed using a radiofrequency (RF)-spoiled echo gradient sequence (FLASH) with the following parameters: flip angle = 30°; echo time = 3.4 ms; TR ≈ 30 ms; resolution = 0.16 × 0.16; number of slices = 3; FOV = 40 × 30 mm; slice thickness = 1 mm; Nex = 10; dynamic scan time ≈ 60 s.

Mice (12 C57BL/6 females; *n* = 3 per condition) were anesthetized by inhalation of isoflurane (1.5%). Each animal then underwent imaging (50 s for each dynamic image) for 8 to 10 min (depending on the variation of the respiration rate). Imaging was stopped and 1 NE formulation (#1 to #4) (see Table 5) was injected intravenously in the tail vein at 53.7 µmol (Fe) kg^−1^ bodyweight. Straight after injection completion, MR image acquisition was restarted and the interval between the injection and the acquisition of the first dynamic image was measured. This interval varied from 1 to 3 min. Finally, the 2 dynamic sessions were fused, considering the injection delay. The time between 2 MR dynamic images for each dynamic session was estimated as the total scan time (which depended on the respiration rate) divided by the number of dynamic images obtained.

For this study, only the signals of the liver and kidneys (renal cortex and renal pyramids) were quantified. To restrict the motion analysis to these organs, a segmentation procedure was performed before analysis. Briefly, on each slice acquired before injection, an initial region of interest (ROI) was defined and subsequently refined by manual correction. Dynamic motion fields were then estimated for each dynamic MR image using the RealTITracker method described by Zachiu et al. [68,69]. These motion (vector) fields were used to update the ROI position in an elastic manner to each dynamic image to follow the organ’s motion throughout the MRI session. The mean MR signals of the liver and one kidney were then extracted from these dynamic ROIs and used to estimate iron accumulation in these organs, as described in the next section. An example of the results of this correction can be found in the Appendix A.

#### 4.7.2. Estimation of the NE Half-Life in the Blood

As the NEs used in this study displayed a longitudinal relaxivity (r_1_) of almost zero [20], the MR signal changes in the liver and kidney over time were estimated to be caused only by $R_{2}^{*}$ changes and not by R_1_ changes. Therefore, the signal equation of the gradient echo sequence (1) was simplified to Equation (2):(1)$$S{\left(\mathrm{TE},\mathrm{TR},\alpha \right)}_{t}=M_{0}\sin\left(\alpha \right)e^{-\mathrm{TE}\cdot R_{2}^{*}}\times \frac{1-e^{-\mathrm{TR}\cdot R_{1}}}{1-\cos\left(\alpha \right)e^{-\mathrm{TR}\cdot R_{1}}}$$
(2)$$S\;\left(t\right)=M_{0}\sin\left(\alpha \right)e^{-\mathrm{TE}\cdot \left(R_{2}^{*}+r_{2}^{*}C\left(t\right)\right)}$$
where α is the flip angle, M_0_ is the MR signal when the echo time (TE) tends to zero, $r_{2}^{*}$ is the transversal relaxivity of the tested NE, and C(t) is the iron concentration originating from the NE sample.

To estimate iron accumulation, the MR signal over time (S_(t)_) was first normalized to the signal measured at the first dynamic acquisition, before NE injection. Because at t = 0, the iron concentration (C(t)) is equal to 0, Equation (2) can be rewritten as:(3)$$S_{0}=M_{0}\sin\left(\alpha \right)e^{-\mathrm{TE}\cdot R_{2\left(0\right)}^{*}}$$
where $R_{2\left(0\right)}^{*}$ is the original relaxation rate. Therefore, the normalized signal of each new dynamic image (S_norm_) is equal to:(4)$$S_{\mathrm{norm}}\left(t\right)=\frac{S\left(t\right)}{S\left(0\right)}=e^{-\mathrm{TE}\cdot r_{2}^{*}C\left(t\right)}$$
and the iron concentration can be estimated over time as:(5)$$C\left(t\right)=\frac{-\ln\left(S_{\mathrm{norm}}\left(t\right)\right)}{\mathrm{TE}\cdot r_{2}^{*}}$$

In vivo, most injected nanoparticles (and also NEs) are cleared from the bloodstream by different organs (liver, spleen, and bone marrow). In this study, the liver was considered to be the main organ of NE clearance and, therefore, iron accumulation in this organ directly reflected blood clearance of NEs. To estimate the apparent NE half-life in the blood, the iron concentration was assumed to change over time as:(6)$$\frac{\mathrm{dC}\left(t\right)}{\mathrm{dt}}=\tau \;(C_{\max}-C\left(t\right))$$
where C_max_ is the maximum iron concentration estimated from the time of NE injection with no surface functionalization and τ is the accumulation rate. Therefore, the clearance rate, τ, was estimated from a Marquardt–Levenberg fit of C(t) using a first-order differential equation (Equation (7)):(7)$$C\left(t\right)=C_{\max}\left(1-e^{-t\tau }\right)$$

Finally, the apparent NE half-life in the blood was defined as:(8)$$t_{1/2\;}=\ln\left(2\right)/\tau$$

#### 4.7.3. In Vivo Dynamic MRI and T_2_* Mapping of Atheromatous Plaques in *Apoe*^−/−^ Mice

Tail vein injections of NE-P3 (#5) were performed at 53.7 µmol(Fe)∙kg^−1^ bodyweight. In one *Apoe*^−/−^ female, the NE half-life in the blood was estimated, based on the FLASH sequence obtained in the dynamic study. Atheromatous plaque labeling by NE-P3 (#5) was monitored in another *Apoe*^−/−^ female by MRI before and at 7 h and 24 h after injection. Isoflurane concentrations were adjusted over time to maintain the respiration rate between 40 and 60 bpm. For in vivo T_2_* mapping of atheromatous plaques, 10 transversal slices passing through the aorta in the thoraco-abdominal region were acquired using a multi-slice RF-spoiled echo gradient sequence: TR/TE1/ΔTE = 1300/2.8/3.6 ms; α = 60°; 15 echoes; NEx = 8; BW = 89.2 MHz; voxel size = 0.1 × 0.1 × 1 mm^3^; FOV = 2.56 × 1.6 cm. Manual segmentation at the aorta level was performed to include only voxels that corresponded to the plaque using the first echo image. The mean T_2_* values were computed for each slice. All calculations were performed using custom scripts written in MATLAB (MathWorks, Natick, MA, USA) and in C++ for the Levenberg–Marquardt algorithm. For each dataset, T_2_* maps were estimated after correction of the macroscopic susceptibilities (e.g., arising from air/tissue interfaces, ΔB0z), as proposed by Dahnke et al. [70]. The limit of acceptability for the signal-to-noise ratio in the T_2_* calculations was fixed to 4. The optical flow algorithm RealTITracker method described by Zachiu et al. [68,69] was used to avoid any image shifts during the echo times that could hamper the correct T_2_* estimate. T_2_* values of >30 ms were not considered to be representative of atheromatous plaques at 4.7 T. These few outliers were excluded from the analysis.

### 4.8. Ex Vivo Analyses of Isolated Aorta Samples

#### 4.8.1. MRI Analyses

Aorta samples embedded in agarose gel (1%) were analyzed ex vivo by MRI using multiple gradient echoes with positive readouts and number of echoes = 20; delta TE = 3 ms; max TE = 59.48 ms; TE/TR = 2.48/100 ms; acquisition bandwidth = 125 kHz; matrix = 256 × 128 × 128; FOV = 20 × 10 × 10; isotropic resolution = 78 µm; number of repetitions = 16; total acquisition time = 7 h 30 min.

#### 4.8.2. Iron Quantification by ESR

ESR experiments were performed at room temperature with a Bruker ESP300E spectrometer operating at X-band frequency (9.54 GHz). The microwave power was set to 20 mW and the magnetic field modulation frequency and amplitude to 100 kHz and 1 mT, respectively. The spectral resolution was 0.7 mT/pt and the acquisition time was 30 min for each sample. After weighing, each aorta sample was carefully digested in HNO_3_ (65%) under a flame. Digestion was repeated 3 times to ensure the sample’s complete mineralization. After the last addition of HNO_3_, 100 mg of NaNO_3_ salt was added to obtain a homogeneous solid powder for the ESR analysis. The recorded ESR spectra were normalized to the aorta mass (expressed in g), and their intensity (obtained by double-integration of the first derivative absorption curve) was compared with that of reference samples with a known concentration of Fe_2_O_3_ nanoparticles (ranging from 5 × 10^−10^ to 1 × 10^−7^ mol/g).

### 4.9. In Vitro Antioxidant Measurement

The antioxidant properties of alpha-tocopherol-containing NEs were assessed by Kirial International/Laboratoires Spiral (Couternon, France) using the KRL test according to Caspar-Bauguil et al. [71]. The antioxidant or prooxidant activity was evaluated in whole blood samples mixed with the different NE formulations and exposed to a controlled free radical attack. The overall resistance of the blood samples against the free radical attack was assessed by calculating the time to produce the hemolysis of 50% of red blood cells (T50% hemolysis, in minutes) after the initiation of the attack. The antioxidant/prooxidant effect of the different NE formulations was then expressed as the percentage of increase/decrease in the T50% hemolysis relative to the control sample (without NE formulation). The results were also expressed in Trolox equivalents (used as a reference).

## 5. Conclusions

HuAbs represent a class of ligands that are theoretically safer for clinical translation. The same is true for NE formulations due to their proven biocompatibility and their easy dissemination through biological barriers. The results of this study pave the way to future non-invasive targeted imaging of atherosclerosis (by combining safe magnetic nanoparticles as a contrast agent and a new anti-galectin-3 HuAb as a targeting agent) and synergistic therapeutic applications using active pharmaceutical ingredients (e.g., alpha-tocopherol, an antioxidant, anti-inflammatory, and cardioprotective vitamin) to induce the regression of the vulnerable plaque. Moreover, a more personalized approach to therapies against atherosclerosis could be guided by molecular imaging. Advances in the in vivo targeting efficiency of agents against proteins overexpressed in the plaque, such as galectin-3, could also improve the assessment and monitoring of atherosclerosis.

## Figures and Tables

**Figure 1 ijms-22-05188-f001:**
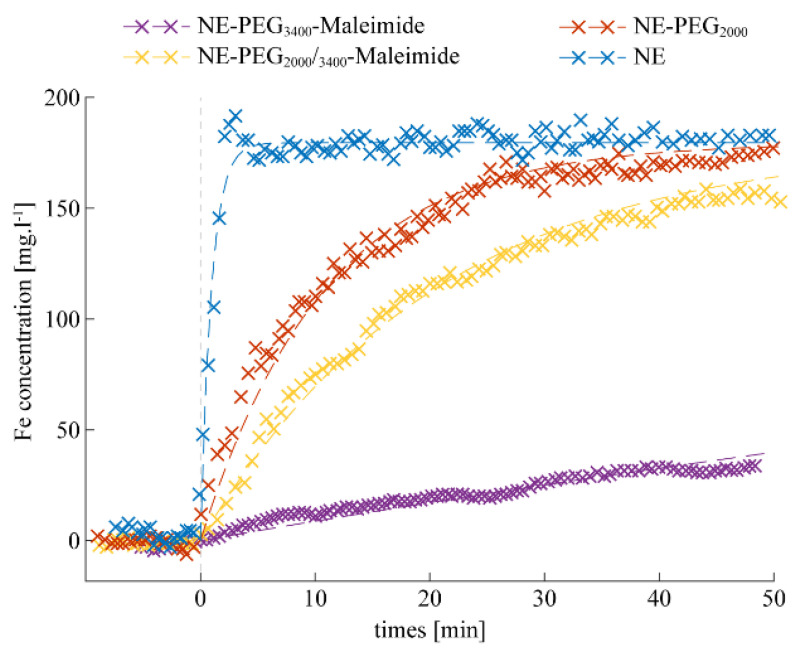
Estimation of the in vivo liver iron (Fe) uptake in mice by dynamic MRI after injection of the different NE formulations. Non-PEGylated NEs (#1) are rapidly cleared from the blood. NE-PEG_3400_-maleimide (#3) displays the best stealth properties with very low liver accumulation at 50 min. NEs decorated with PEG_2000_ (#2) or PEG_2000_/PEG_3400_–maleimide (#4) displayed a similar profile, with a relatively rapid clearance. Each NE half-life was estimated from the corresponding graph (dashed lines in Figure 1; Table 2).

**Figure 2 ijms-22-05188-f002:**
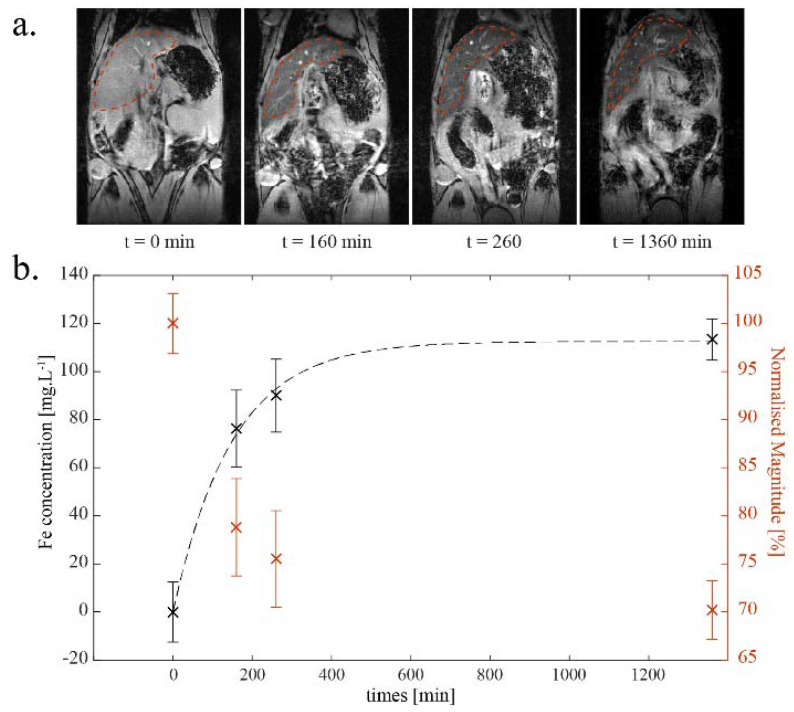
NE-P3 (#5) half-life in the bloodstream was determined by dynamic MRI. (**a**): MR images at different time points showing the regions of interest (ROI, inside the red dashed lines) that encompass the liver and were used to measure the mean magnitude of the signal. (**b**): Mean normalized magnitude of the signal (red) and the corresponding iron concentration (black) estimated from the transverse relaxivity $r_{2}^{*}$ of NE- P3 (#5). Error bars come from the normalized magnitude’s standard deviation of the voxels included in each ROI. The conjugation of biological moieties (scFv-Fc-2Cys P3 HuAb) to the PEGylated NE surface decreased the NE’s half-life in the blood from 139 min to 103 min (calculated from the graph of the iron concentration, dashed line).

**Figure 3 ijms-22-05188-f003:**
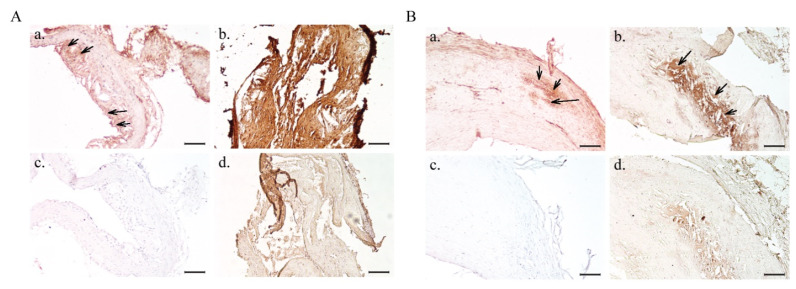
IHC analysis of aorta sections from hypercholesterolemic *Apoe*^−/−^ mice (**A**) and human endarterectomy samples (**B**). NE-P3 (#5) can recognize galectin-3 in mouse aorta sections (**Ab**) and in human samples (**Bb**). No signal was observed with NE–PEG_3400_–maleimide (#3) (without antibody) (**Ad**,**Bd**) and in the negative control (secondary antibody only: **Ac**,**Bc**). Unconjugated scFv-Fc-2Cys P3 (**Aa**,**Ba**) was used as a positive control. Black arrows highlight specific binding by P3 or NE-P3 (#5). Scale bar: 250 µm (upper panels) and 100 µm (lower panels). Two tissue blocks were used for mouse aorta sections labeling in Panel A (one for a and c, and another for b and d). The same tissue block from the same patient was used in Panel B. The mouse aorta and endarterectomy sections shown are representative images taken from successive stained sections. Three independent experiments were performed with these formulations (a section per formulation).

**Figure 4 ijms-22-05188-f004:**
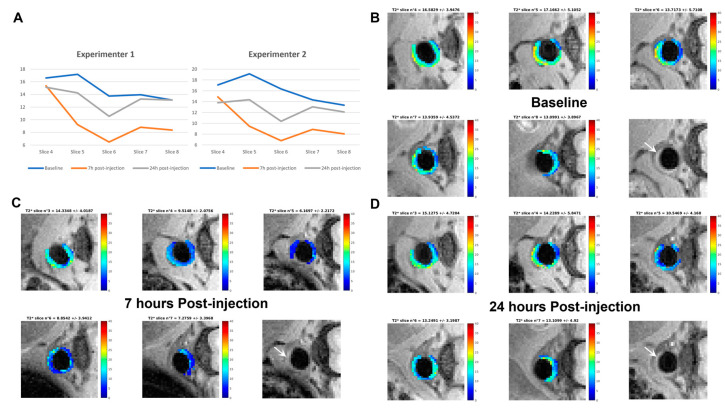
In vivo atheroma T_2_* variations after injection of NE-P3 (#5). (**A**): Diagrams made by two experimenters representing the T_2_* mean values (in ms) for each slice of the abdominal aorta of one *Apoe*^−/−^ mouse at baseline, 7 h, and 24 h after NE-P3 (#5) injection. Typical segmented T_2_* maps of five slices before (**B**), 7 h (**C**), and 24 h (**D**) after NE-P3 (#5) injection, shown with the color scale. Slices 4, 5, 6, 7, and 8 before injection correspond to Slices 3, 4, 5, 6, and 7 after injection. After injection, great care was taken to reposition the mouse in exactly the same way as before injection; however, sections before and after injection could be shifted by one or two slices. The corresponding magnitude image (first echo) of the third slice in each dataset (i.e., Slice 6 at baseline and Slice 5 post-injection) is shown in the rightmost lower panel at each time point. In this image, the atheromatous plaque is clearly visible in white (arrow). Slice-by-slice T_2_* maps were obtained using a homemade MATLAB-based tool.

**Figure 5 ijms-22-05188-f005:**
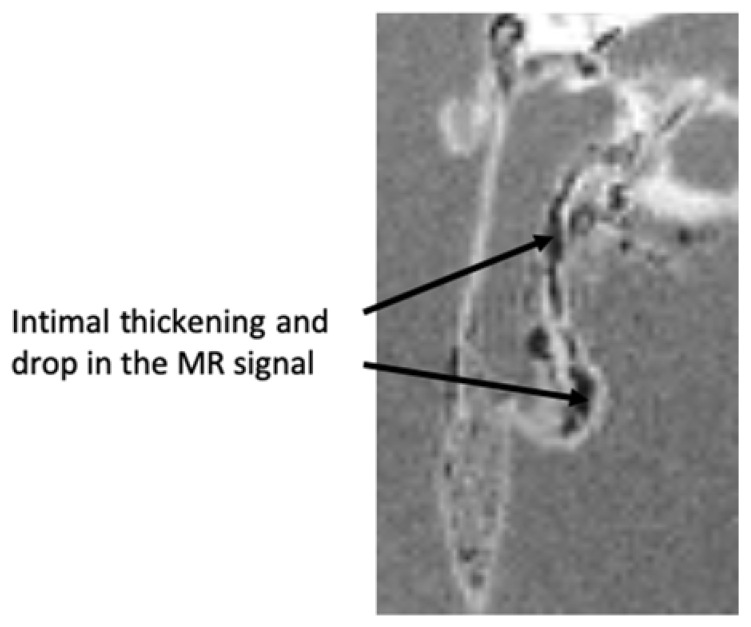
MRI magnitude image of the agarose-embedded aorta after isolation from one *Apoe*^−/−^ mouse following in vivo injection of NE-P3 (#5) in the tail vein. Black arrows highlight the decrease in the MR signal.

**Figure 6 ijms-22-05188-f006:**
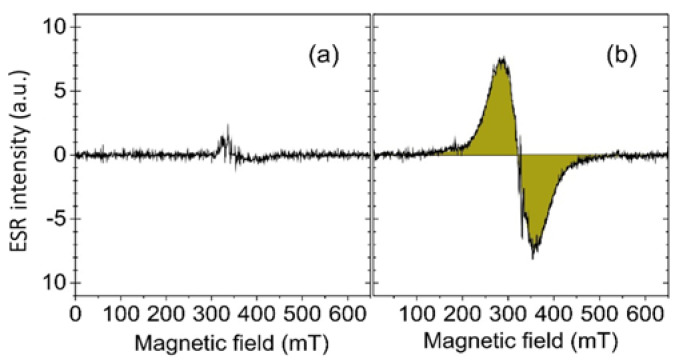
Room temperature X-band (9.54 GHz) Electron Spin Resonance (ESR) spectra of (**a**) an aorta isolated from a *Apoe*^−/−^ mouse untreated (control) and (**b**) from a *Apoe*^−/−^ mouse after NE-P3 (#5) injection.

**Figure 7 ijms-22-05188-f007:**
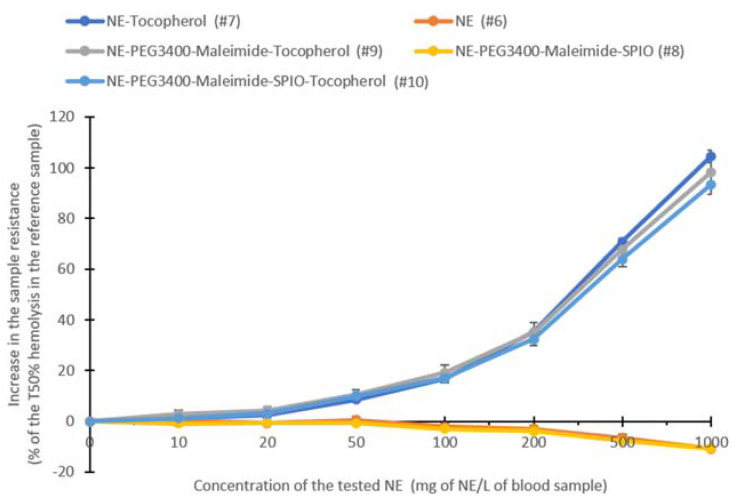
“Kit Radicaux Libres” test (KRL test) to determine the antioxidant properties of the tested NE formulations (Table 3). The graph shows the change (in percent) of the time in which 50% of red blood cells were lysed (T50% hemolysis in minutes) after the initiation of free radical attack relative to the control sample (*n* = 3).

**Figure 8 ijms-22-05188-f008:**
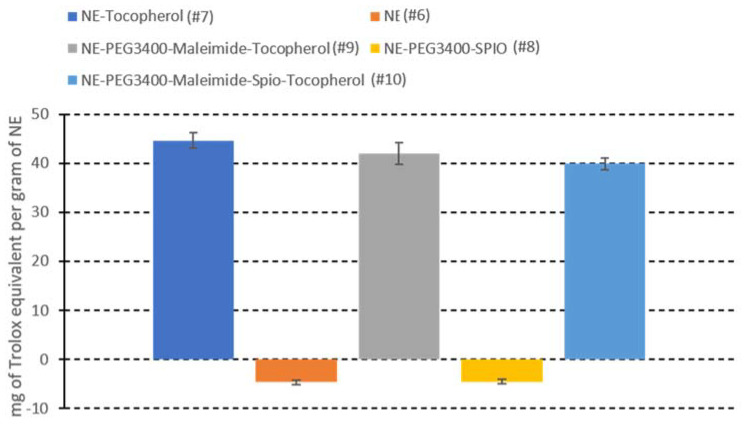
Antioxidant properties of the tested NE formulations expressed as mg of Trolox equivalent per gram of NE (*n* = 3).

**Table 1 ijms-22-05188-t001:** NEs’ physicochemical data.

	NE	NE–PEG_2000_	NE–PEG_3400_–Maleimide	NE–PEG_2000/3400_–Maleimide
Formulation Number	(#1)	(#2)	(#3)	(#4)
Mean diameter (nm) ± SD (*n* = 3)	175.8 ± 1.0	190.9 ± 2.2	197.2 ± 4.6	191.0 ± 2.4
Polydispersity index	0.108	0.134	0.115	0.097

NE: nano-emulsion; PEG: polyethylene glycol; SD: standard deviation.

**Table 2 ijms-22-05188-t002:** Half-life of the tested NE formulations in the bloodstream.

	NE	NE–PEG_2000_	NE–PEG_3400_–Maleimide	NE–PEG_2000/3400_–Maleimide	NE–PEG_3400_–Maleimide-P3
Formulation Number	(#1)	(#2)	(#3)	(#4)	(#5)
Blood half-life (min) ± SD (*n* = 3) * (*n* = 1)	<1	7.2 ± 3.4	14.1 ± 2.5	138.7 ± 11.2	103 *

**Table 3 ijms-22-05188-t003:** NE composition for the antioxidant assessment.

	NE	NE–Tocopherol	NE–PEG_3400_–Maleimide–SPIO	NE–PEG_3400_–Maleimide–Tocopherol	NE–PEG_3400_–Maleimide–SPIO–Tocopherol
Formulation Number	(#6)	(#7)	(#8)	(#9)	(#10)
Miglyol 840 (%)	20	15	20	15	15
Alpha-tocopherol (%)	-	5	-	5	5
Tween 80 (%)	2.5	2.5	2.5	2.5	2.5
Lipoid 80 (%)	1.2	1.2	1.2	1.2	1.2
Milli-Q water (%)	qs 100	qs 100	qs 100	qs 100	qs 100
SPIO (µmol/mL)	-	-	12.5	-	12.5
Lipid–PEG_3400_–maleimide (µmol/mL)	-	-	4.4	4.4	4.4

SPIO: superparamagnetic iron oxide nanoparticles; qs: quantum satis.

**Table 4 ijms-22-05188-t004:** NE characterization.

	NE	NE–Tocopherol	NE–PEG_3400_–Maleimide–SPIO	NE–PEG_3400_–Maleimide–Tocopherol	NE–PEG_3400_–Maleimide–SPIO–Tocopherol
Formulation Number	(#6)	(#7)	(#8)	(#9)	(#10)
Mean diameter (nm) ± SD (*n* = 3)	173.5 ± 0.8	171.7 ± 0.5	162.6 ± 2	181.3 ± 1.8	213.7 ± 1.2
Polydispersity index	0.140	0.152	0.123	0.163	0.255

**Table 5 ijms-22-05188-t005:** NE composition.

	NE	NE–PEG_2000_	NE–PEG_3400_–Maleimide	NE–PEG_2000/3400_–Maleimide	NE–PEG_3400_–Maleimide–P3
Formulation Number	(#1)	(#2)	(#3)	(#4)	(#5: NE-P3)
Miglyol 840 (%)	20	20	20	20	20
Tween 80 (%)	2.5	2.5	2.5	2.5	2.5
Lipoid 80 (%)	1.2	1.2	1.2	1.2	1.2
Milli-Q water (%)	qs 100	qs 100	qs 100	qs 100	qs 100
SPIO (µmol/mL)	12.5	12.5	12.5	12.5	12.5
Lipid–PEG_2000_ (µmol/mL)	-	5	-	3.8	-
Lipid–PEG_3400_–maleimide (µmol/mL)	-	-	5	1.2	5
P3 antibody (nmol/mL)	-	-	-	-	1.9

## Data Availability

The data presented in this study are available on request from the corresponding author.

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
