# Peer review of "A Nano-Emulsion Platform Functionalized with a Fully Human scFv-Fc Antibody for Atheroma Targeting: Towards a Theranostic Approach to Atherosclerosis"

_ijms, 2021, doi:10.3390/ijms22105188_

Round 1
Reviewer 1 Report
Atherosclerosis is at the onset of cardiovascular diseases that are among the leading causes 17 of death worldwide. Currently, high-risk plaques, also called vulnerable atheromatous plaques, 18 remain often undiagnosed until the occurrence of severe complications, such as stroke or 19 myocardial infarction. Molecular imaging agents that target high-risk atheromatous lesions could 20 greatly improve the diagnosis of atherosclerosis by identifying sites of high disease activity. 21 Moreover, a “theranostic approach” that combines molecular imaging agents (for diagnosis) and 22 therapeutic molecules would be of great value for the local management of atheromatous plaques. 23 The aim of this study was to develop and characterize an innovative theranostic tool for 24 atherosclerosis. We engineered oil-in-water nano-emulsions (NEs) loaded with superparamagnetic 25 iron oxide (SPIO) nanoparticles for magnetic resonance imaging (MRI) purpose. Dynamic MRI 26 showed that NE-SPIO nanoparticles decorated with a polyethylene glycol (PEG) layer reduced their 27 liver uptake and extended their half-life. Then, the NE-SPIO-PEG formulation was functionalized 28 with a fully human scFv-Fc antibody (P3) recognizing galectin 3, an atherosclerosis biomarker. The 29 P3-functionalized formulation targeted atheromatous plaques as demonstrated in 30 immunohistochemistry analyses of mouse aorta and human artery sections and in an Apoe-/- mouse 31 model of atherosclerosis. Moreover, the formulation was loaded with SPIO nanoparticles and/or 32 alpha-tocopherol to be used as a theranostic tool for atherosclerosis imaging (SPIO) and for delivery 33 of drugs that reduce oxidation (here, alpha-tocopherol) in atheromatous plaque. This study paves 34 the way to non-invasive targeted imaging of atherosclerosis and synergistic therapeutic 35 applications (abstract provided by the author).
This is a very well written paper. The reviewer has two suggestions to improve the paper.
1. Current view of vulnerable plaque in clinical setting
Vulnerable plaque has been sought as a culprit of acute coronary syndrome and numerous RCTs among patients with stable ischemic heart disease tested this hypothesis comparing PCI (and other intervention) and intensive medical therapy, including COURAGE, BARI2D and recent ISCHEMIA trial. In fact, a recent systematic review and meta-analysis 1 showed that there is no survival benefit of revascularization among patients with stable ischemic heart disease. One potential explanation for this phenomenon is a vast majority of ‘vulnerable plaque’ is actually healed.2 Several investigators suggested that risk would be more strongly predicted by using total atheromatous burden of the coronary artery,3 such as coronary calcium score. This standpoint needs to be incorporated into the discussion.
2. α tocopherol
Antioxidant has been suggested to prevent many non-communicable chronic diseases including atherosclerosis. However, RCTs of supplementation of antioxidant on cardiovascular disease failed to show its benefit. Is there any evidence to show to support that a theranostic delivery of α tocopherol stabilizes vulnerable plaque?
- Bangalore S, Maron DJ, Stone GW, Hochman JS. Routine Revascularization Versus Initial Medical Therapy for Stable Ischemic Heart Disease: A Systematic Review and Meta-Analysis of Randomized Trials. Circulation. 2020;142(9):841-857.
- Stone GW, Maehara A, Lansky AJ, et al. A Prospective Natural-History Study of Coronary Atherosclerosis. New England Journal of Medicine. 2011;364(3):226-235.
- Arbab-Zadeh A, Fuster V. From Detecting the Vulnerable Plaque to Managing the Vulnerable Patient: JACC State-of-the-Art Review. J Am Coll Cardiol. 2019;74(12):1582-1593.
Author Response
We really thank the reviewers for their very valuable comments they made on our paper. We corrected and improved the manuscript following their suggestions. We also provide a point-by-point response to the reviewer's comments (please see the attachement).
We hope that all these replies and the uploaded version of our paper will satisfy the reviewers.
Again, thank you so much for the help you provide us into the improvement of this work.

Reviewer 2 Report
Summary
Authors Bonnet et al are presenting a path toward a theranostic for specific MRI-based imaging and treatment to diagnose and control vulnerable atheromatous plaques in atherosclerosis. In their presentation the authors following a rational step-wise approach by first demonstrating half-life and biodistribution improvements of pegylated NE-SPIO nanoparticles, then functionalizing the pegylated NE-SPIO with a human antibody fragment construct to specifically target unstable plaques and finally characterizing NE-SPIO formulation with an antioxidant. Throughout the in-vitro and pre-clinical in-vivo work, authors highlight suitability for clinical use.
Broad comments
The authors Marie-Josée Jacobin-Valat, , Martine Duonor-Cérutti, Gisèle Clofent-Sanchez and Audrey Hémadou are listed as inventors on the patent for the human anti-galectin-3 antibody (WO2019068863A1, https://patents.google.com/patent/WO2019068863A1/en) used in this work. The authors reference the patent in the main text (line 64 and 331). It should also be added to the conflict of interest statement (line 631-632)
The title suggests the development of an antibody-functionalized theranostic. The formulation characterized in paragraph 2.7, although containing maleimide and being suitable to be conjugated to the scFvP3 construct, is not functionalized with the human Ab yet. Please elaborate on this point (future studies etc).
The title mentions this work being envisioned as a 'platform' by the authors. Is this in regards to various payloads (other than alpha-tocopherol) or the possible use of alternative human Abs? Please discuss.
Specific comments
Page 2, line 66
Would suggest to include recent work from other groups regarding SPION-loaded NEs as contrast agent as well (e.g. Wallyn J, Anton N, Mertz D, Begin-Colin S, Perton F, Serra CA, Franconi F, Lemaire L, Chiper M, Libouban H, Messaddeq N, Anton H, Vandamme TF. Magnetite- and Iodine-Containing Nanoemulsion as a Dual Modal Contrast Agent for X-ray/Magnetic Resonance Imaging. ACS Appl Mater Interfaces. 2019 Jan 9;11(1):403-416. doi: 10.1021/acsami.8b19517. Epub 2018 Dec 26. PMID: 30541280.)
.
Line 66 - 69
Emphasizing the unique role of Trem2-high macrophages in the plaque but not the healthy aorta would further strengthen the choice of the huAb 3 (Cochain C, Vafadarnejad E, Arampatzi P, et al Single‐cell RNA‐seq reveals the transcriptional landscape and heterogeneity of aortic macrophages in murine atherosclerosis. Circ Res 2018; 122: 1661–1674.)
Line 126
Please explain your dosing choice of 3mg/kg
Figure 3
The inclusion of human tissues in the IHC analysis is a great gateway toward discussing of a clinical application. Please specify whether sections shown here are taken from the same tissue block/same patient. Please also specify whether images shown in this figure are representative images taken from multiple stained section or a single stained section used for qualitative illustration. Staining and annotations are clear and convincing and are very well explained in the text.
The images for the mouse are well explained in the text. Please again specify if the sections used here for staining and controls are taken from the same tissue sample and whether they are representative images for a series of stained sections.
Paragraph 2.6.2 and figure 6
Please clarify if only one sample per group was analyzed as in paragraph 2.6.1/figure 5 or if this is a representative ESR spectrum from multiple samples per group.
Paragraph 2.7
Please briefly reason the choice of tocopherol in this section or introduction.
Line 367-371
Very good point to bring up importance of toxicity as prerequisite for clinical development of the NE-SPIO tocopherol theranostic. This point however would need some clarification. Would 3mg/kg be the projected clinical dose or would this be an excessive dose used in a tox study? Why? Please further explain why you would expect a similar toxicity profile as ferumoxytol (route of delivery, half-life etc).
Throughout the manuscript:
Please specify error-bars depicted in graphs in the figure legend (and or y-axis title) by clarifying which analysis has been used and number of replicates. Also please specify meaning of ranges (+/-) in text and tables (e.g. Table 1, line 115)
Throughout the manuscript
Please doublecheck abbreviations and initialisms throughout the text and figure legends. Make sure explanation is given at the first mentioning in the main text and used throughout. If term is used only once or twice in the text, no abbreviation would be necessary.
Throughout the manuscript
NIST (National Institute of Standards and Technology) recommends the use of capital 'L' for 'liter'. Please be at least consistent throughout the text and figures.
Author Response

(The authors gave the same response as above.)
